## Review Article

**Author for correspondence:**
Alexander M. Jones,
E-mail: alexander.jones@slcu.cam.ac.uk

# Focus on biosensors: Looking through the lens of quantitative biology

## James H. Rowe and Alexander M. Jones

Sainsbury Laboratory, Cambridge University, Cambridge, United Kingdom

### Abstract

In recent years, plant biologists interested in quantifying molecules and molecular events in vivo have started to complement reporter systems with genetically encoded fluorescent biosensors (GEFBs) that directly sense an analyte. Such biosensors can allow measurements at the level of individual cells and over time. This information is proving valuable to mathematical modellers interested in representing biological phenomena in silico, because improved measurements can guide improved model construction and model parametrisation. Advances in synthetic biology have accelerated the pace of biosensor development, and the simultaneous expression of spectrally compatible biosensors now allows quantification of multiple nodes in signalling networks. For biosensors that directly respond to stimuli, targeting to specific cellular compartments allows the observation of differential accumulation of analytes in distinct organelles, bringing insights to reactive oxygen species/calcium signalling and photosynthesis research. In conjunction with improved image analysis methods, advances in biosensor imaging can help close the loop between experimentation and mathematical modelling.

## 1. The first glimmers of green light

Since the cryptic intron was removed from *Aequorea victoria* green fluorescent protein (GFP; Haseloff et al., 1997), fluorescent protein (FP) expression has had an incalculable impact on plant biology. The diverse uses of FPs include characterising protein expression pattern and localisation (Benková et al., 2003), marking the endomembrane system (Geldner et al., 2009), demonstrating intracellular transcription factor movement (Nakajima et al., 2001; Raissig et al., 2017) and quantifying phloem unloading (Stadler et al., 2004). Some of the most innovative FP uses have come from the development of GEFBs. These are proteins that contain one or more FPs, the fluorescent properties of which change with a given stimulus. This outlook piece will not detail the many biosensors used in plants [for a comprehensive overview, see a recent review (Walia et al., 2018)] but instead will emphasise recent developments, particularly of GEFBs that report on molecules and molecular events via direct sensing and independently of other components. We will then explore how biosensors are leading to a deeper more quantitative analysis of plant biology by providing more robust data for model construction and parametrisation.

## 2. From transcriptional reporters to direct biosensors

Over the years, FP expression has been placed under the control of a variety of promoters, many of which were chosen for their responsiveness to a given stimulus, such as a hormone. This principle was taken further with the development of synthetic promoters, which act as indirect hormone reporters, such as the auxin responsive DR5 promoter. In DR5::GFP, tandem repeats of an auxin responsive element are placed upstream of a minimal 35S Cauliflower Mosaic Virus promoter, to drive expression of GFP (Figure 1a; Benková et al., 2003; Ulmasov et al., 1997). Higher FP fluorescence in a cell or tissue indicates sites of increased auxin response. Using fluorescent transcriptional reporters to infer hormone accumulation revolutionised plant developmental biology and acted as the basis for a series of spatial models of hormone distributions, helping establish the developmental regulation of patterning by auxin in both shoots and roots (Grieneisen et al., 2007; Jönsson et al., 2006).

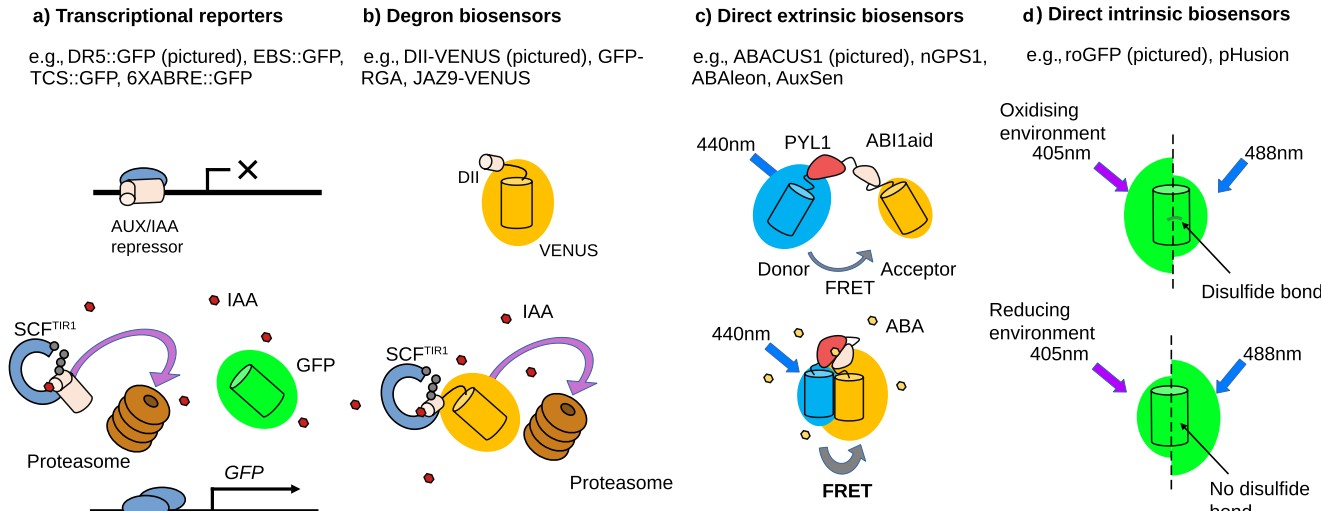

**a) Transcriptional reporters**

e.g., DR5::GFP (pictured), EBS::GFP, TCS::GFP, 6XABRE::GFP

**b) Degron biosensors**

e.g., DII-VENUS (pictured), GFP-RGA, JAZ9-VENUS

**c) Direct extrinsic biosensors**

e.g., ABACUS1 (pictured), nGPS1, ABAleon, AuxSen

**d) Direct intrinsic biosensors**

e.g., roGFP (pictured), pHusion

**Fig. 1.** Different types of genetically encoded fluorescent biosensors and their molecular mechanisms. (a) The mechanism of action of the auxin reporter DR5::GFP, where the auxin receptor complex targets the AUX/IAA transcriptional repressors for degradation, allowing green fluorescent protein transcription and fluorescence in the presence of the hormone. (b) The mechanism of action of the auxin biosensor DII:VENUS, where a venus fluorescent protein (FP) fused to domain II (DII) from an AUX/IAA protein is targeted for degradation by the auxin receptor complex, causing loss of fluorescence in the presence of the hormone. (c) The mechanism of action of the direct abscisic acid biosensor ABACUS1, where hormone binding causes the interaction of two sensory domains of a single fusion protein. This conformational change causes increased Förster resonance energy transfer (FRET) between two FP domains (a FRET donor and acceptor) and an altered emission ratio upon donor excitation. (d) The mechanism of action of the direct glutathione redox potential biosensor roGFP. A disulphide bond on the barrel of roGFP is reversibly sensitive to oxidation, causing an altered absorption spectrum.

This approach has been extremely useful and has consequently been used to create reporters for other hormones such as cytokinin and abscisic acid (ABA; Wu et al., 2018; Zürcher et al., 2013), but at best transcriptional reporters such as DR5 indicate where hormone responses were. The levels and distribution of plant hormones are dynamically adjusted, to regulate a variety of processes such as tropisms (Band et al., 2014), developmental patterning (Benková et al., 2003) and stress responses (Iuchi et al., 2000; Rowe et al., 2016). The time required for transcription, translation and maturation of FPs (Balleza et al., 2018) mean that fluorescence increases following stimuli are delayed; in fact, a DR5::VENUS auxin response does not become visible until 2 hr after the mRNA accumulation and follows different dynamics (Brunoud et al., 2012). FP longevity is such that a reduction in fluorescence following removal of a stimulus could be even slower, for example, wildtype GFP has a 26-hr half-life in mammalian systems (Corish & Tyler-Smith, 1999). Transcriptional reporters are indirect and so rely on endogenous signalling components and transcriptional/translational machinery, which may vary over time or from tissue to tissue (Bargmann et al., 2013; Prigge et al., 2020).

The next big advance in imaging hormone responses came with the advent of degradation-based reporters such as DII-VENUS. The second domain (DII) from IAA28 was fused to the yellow FP Venus, to create a fluorescent fusion protein that is ubiquitinated by the SCF^TIR1 complex upon auxin perception (Figure 1b; Brunoud et al., 2012). DII-VENUS ubiquitination causes rapid degradation and loss of fluorescence, reducing the time delay between hormone accumulation and response (Brunoud et al., 2012). This facilitated the modelling of fast root auxin redistribution due to gravistimulation (Band et al., 2014). Whilst this is a great improvement in imaging hormone accumulations, problems arise in determining whether differences in fluorescence are due to changes in expression or degradation of the FP. A powerful tool to overcome this problem is the expression of a second FP, as in the R2D2 or qDII reporters (Galvan-Ampudia et al., 2020; Liao et al., 2015). By comparing the DII-VENUS fluorescence, which is lost in the

presence of auxin, to a stabilised control FP, tissue specific expression and optical differences are taken into account, leading to better estimates of auxin concentration. However, DII-VENUS, R2D2 and qDII still rely on the endogenous SCF^TIR1 and proteasomal machinery to elicit fluorescence changes, so could experience tissue to tissue variation in response, and they cannot be used to image fast hormone depletions or oscillations as VENUS maturation can take upwards of 18 min in bacterial systems (Balleza et al., 2018).

One approach to overcome these problems is to engineer a fluorescent biosensor that changes its fluorescence properties depending on direct interaction with the molecule or molecular event of interest and independently of other components; these are the direct GEFBs. Many also have a ratiometric output to control for optical artefacts (such as the loss of signal in deeper tissues) during imaging and differences in expression.

Whilst direct GEFBs for small second messengers, such as calcium, have existed for a long time (Miyawaki et al., 1997), the first direct plant hormone GEFBs were developed fairly recently and all follow a similar design principle to previous ratiometric Förster resonance energy transfer (FRET) sensors (Herud-Sikimić et al., 2021; Jones et al., 2014; Rizza et al., 2017; Waadt et al., 2014). The abscisic acid concentration and uptake sensor 1-2μ (ABACUS1-2μ), for example, consists of a sensory domain and two FPs, all connected by linkers to form a single protein (Figure 1c). The FPs undergo FRET from the donor edCerulean to the acceptor edCitrine upon donor excitation. The amount of energy transfer is sensitive to the distance and orientation of the FPs (Jares-Erijman & Jovin, 2003). Upon ABA binding, the biosensor undergoes a conformational change, which causes changes in FRET and thereby in FP emission ratio (Jones et al., 2014). ABACUS1-2μ is an example of an extrinsic sensor, as it relies on a sensory domain separate to the fluorescence domain to respond to a stimulus.

Intrinsic sensors, where the fluorescence properties of the FP itself are varied by direct interactions with the stimulus, also exist (Walia et al., 2018). roGFP, for example, is an intrinsic ratiometric sensor for the glutathione redox potential (E$_{GSH}$; Figure 1d; Hanson

et al., 2004). A disulphide bond on the barrel of roGFP is reversibly sensitive to oxidation, causing an altered absorption spectrum. By sequentially exciting the FP at two different wavelengths, and quantifying emission, a ratio of the oxidised and reduced roGFP is obtained, which indicates $E_{GSH}$ at the subcellular level (Hanson et al., 2004; Jiang et al., 2006).

There are too many other sensor types and designs to be summarised here, and sensor development is the subject of much ongoing research (Walia et al., 2018). An ideal sensor is reversible, ratiometric, specific, responsive to the analyte in the endogenous range, has minimal impact on endogenous signalling (i.e., highly orthogonal) and therefore minimal phenotypes and has a large signal-to-noise ratio. Developing sensors that fulfil all of these criteria can take decades, but even imperfect sensors can be useful in a host of research contexts. As numerous sensors for diverse analytes are now widely available, researchers can simultaneously look at different steps in complex systems to quantitatively understand signal processing at high spatiotemporal resolution as detailed in the next section.

## 3. Sensing signalling in time and space

A biological system will often contain interacting multireaction pathways, which vary the concentration of a host of chemical species. When representing a system in a model, choosing which interactions to include and exclude is a difficult process that strongly affects the model behaviour and predictions (Aldridge et al., 2006). Models are often simplified for a lack of information or for computational reasons, as reducing the number of unconstrained parameters vastly reduces the computation required for parameter scanning. A common simplification is to reduce a multistep signalling pathway to single reaction, with the assumption that this will not affect the model outcomes, but recent work has shown that this often sacrifices the ability of the model to reproduce temporal dynamics (Korsbo & Jönsson, 2020). Using transcriptional reporter systems, the experimental outputs are abstracted from the signalling by multiple steps, so the temporal dynamics have been difficult to measure, making model validation difficult.

But for many signalling networks, there are now direct GEFBs for multiple steps, which can be tracked in real time in living cells, allowing the network to be dissected in fine detail. This is exemplified in abiotic stress responses which often involve responses in calcium, ABA, Sucrose nonfermenting-1-related protein kinase 2 activity and reactive oxygen species, which may show differing yet overlapping dynamics to regulate processes such as gene expression and stomatal closure. Each of these signalling steps now also has a suite of direct biosensors (Huang et al., 2019; Jones et al., 2014; Waadt et al., 2014; Zhang et al., 2020), allowing high spatiotemporal quantification.

Recent pioneering co-expression of pairs of direct biosensors has even allowed the temporal dynamics of many second messengers or hormones to be characterised simultaneously (ABA and $Ca^{2+}$; pH and $E_{GSH}$; pH and $H_2O_2$; $Ca^{2+}$ and $E_{GSH}$; $Ca^{2+}$ and H2O2; $Ca^{2+}$, pH and $Cl^-$ (Keinath et al., 2015; Waadt et al., 2017; 2020). This has confirmed many predictions and offered new insights. Whereas auxin application caused the fast modulation $Ca^{2+}$ and $H^+$ dynamics in roots, ABA application caused no such fast ion dynamics. This response stands in contrast with stomata, where ABA microinjection elicits a $Ca^{2+}$ response in the majority of stomata, enhancing closure (Huang et al., 2019; Waadt et al., 2017).

Such detailed datasets will no doubt be invaluable in constructing realistic models of signal integration in the future.

In addition to temporal responses, spatial signalling responses can also be measured with biosensors. Unlike transcriptional/degron-based reporters, direct GEFBs that independently sense their molecule or molecular event of interest allow the subcellular dynamics of signalling systems to be assessed. This has revealed the spatial importance of subcellular localisation of calcium accumulation in root growth (Leitão et al., 2019) and the differing redox status of different organelles (Exposito-Rodriguez et al., 2017; Schwarzländer et al., 2008). Most biosensors so far have focused on the nuclear or cytoplasmic space, but as experimentalists are targeting biosensors to other cellular compartments, so we will see a whole host of new, spatially distinct responses in the coming years, allowing the modelling and understanding of interorganelle signalling.

Unfortunately, many biosensors currently in use contain FPs that might be unsuitable for highly oxidising or acidic compartments present in some organelles or the apoplasm, although changing the FPs may help address this (Erard & Guiot, 2015). The FP engineering community has constantly strived to produce a diverse pallet of FPs that are now listed in FPbase, providing an invaluable tool for biosensor engineering. FPbase is a repository, which consolidates the fluorescence spectra, quantum yield, acid sensitivity (i.e., pKa), key mutations and other important properties for a vast number of FPs in a searchable format (Lambert, 2019). When combined with increasingly affordable nucleic acid synthesis technology, high throughput combinatorial cloning techniques provide a powerful platform for rapid biosensor engineering. The expanding diversity of GEFBs and their applications has necessitated improved analysis methods to extract and interpret diverse datasets.

## 4. Image analysis and data processing, overcoming a major bottleneck to interpreting large datasets

In recent years, there has been a renaissance in development of imaging modalities and analysis methods. For analysis, simple tools like ImageJ have been expanded into flexible toolsets like Fiji (Fiji Is Just ImageJ), with multitudes of user-written plug-ins (Schindelin et al., 2012), and the use of comprehensive analysis suites, such as Morphographx/LithographX (de Reuille et al., 2015) or IMARIS (https://imaris.oxinst.com/), is now commonplace, offering compelling nearly complete segmentations and reconstructions of entire organs.

The first step in image analysis is often segmentation (Figure 2), the process of deciding which parts of an image are of interest for analysis and labelling them so data can be extracted (Long et al., 2012). At the simplest level, this involves user-drawn regions of interest (ROIs) in tools like ImageJ, whilst more complex analysis may involve automated/semi-automated segmentation performed computationally.

The importance of a good segmentation becomes apparent in the subsequent quantification steps. For intensiometric sensors, this involves quantifying one signal channel, whereas for ratiometric sensors, this may involve dividing the signal from one channel by another (Mahlandt & Goedhart, 2021; Rizza et al., 2019). For ratiometric sensors, quantifying fluorescence emission in areas with poor/no signal can lead to artefactually large variation and a low signal-to-noise ratio as can including pixels where the detector

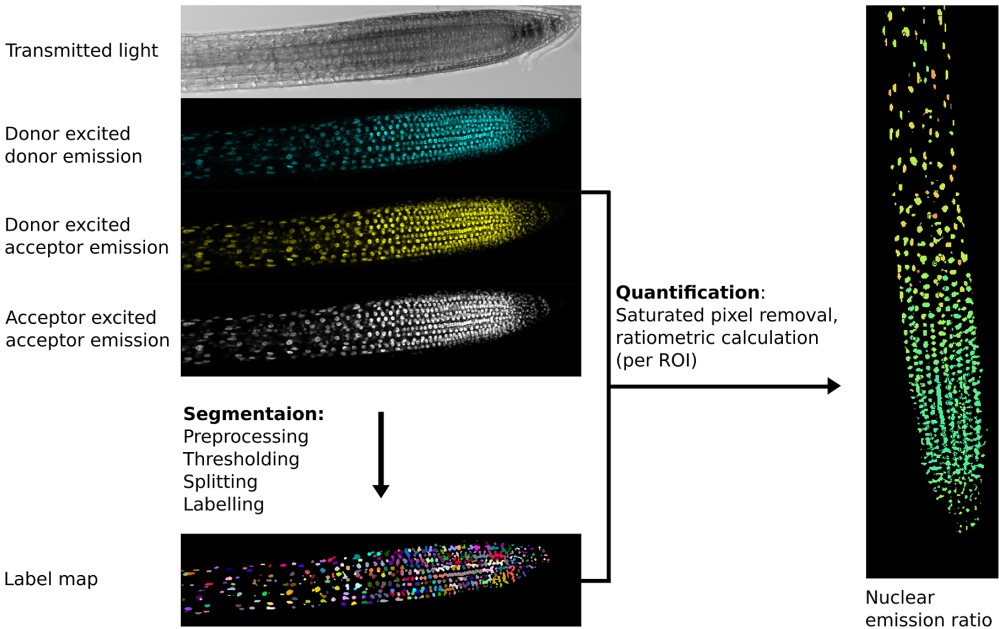

Transmitted light

Donor excited
donor emission

Donor excited
acceptor emission

Acceptor excited
acceptor emission

**Segmentaion:**
Preprocessing
Thresholding
Splitting
Labelling

**Quantification:**
Saturated pixel removal,
ratiometric calculation
(per ROI)

Label map

Nuclear
emission ratio

**Fig. 2.** Example image processing pipeline for the nuclear localised ratiometric Förster resonance energy transfer sensor nlsABACUS1-2μ.
*Notes:* Several image acquisition channels are used in FRET biosensor analysis. Here, the new FRETENATOR analysis pipeline for Fiji was used (Rowe et al., 2021). For segmentation, the following steps are performed: first a difference of Gaussian filter is applied to the acceptor-excited acceptor-emission channel to remove background and smooth noise, then Otsu's method is used to threshold the filtered image based on signal intensity. A watershed algorithm is used to split touching objects in the threshold image, and a connected-component analysis is used to label all the individual objects, producing a label map. The quantification steps involve removing saturated voxels from the original images and then dividing the mean signal intensity of donor-excited acceptor-emission channel by the donor-exited donor-emission channel for each labelled ROI, which can be represented on the segmented image using false coloration (e.g., the Turbo lookup table used here).

is saturated—such areas should be removed prior to interpretation (Figure 2).

Focal drift, sample movement and growth also present problems for time course imaging and analysis. Often depth ($z$-plane) movement causes brightness changes (an optical artefact), which is a particular problem for intensiometric GEFBs. During analysis, if ROIs fail to move in step with the moving object, then additional artefacts may be introduced. Imaging samples in microfluidics systems to limit sample movement can help eliminate some of these issues, but often they are dealt with during the analysis stage. One solution is to identify features that move and offset each timeframe in an image sequence to eliminate movement (image registration and translation). If object displacement is not uniform, such as in a growing tissue, then tracking of segmented objects may help address any artefacts from movement.

Biosensor images present further challenges for analysis, as they often require multichannel imaging and sequential acquisition, which must be taken at high speed when measuring fast dynamics. Because of the complexity of the imaging setup, a compromise is often made between acquisition speed, resolution and signal-to-noise ratio (Rizza et al., 2019; Rowe et al., 2021). This compromise means accurate fully automated segmentation and complete tissue reconstruction is difficult, but the development of deep learning segmentation tools, such as PlantSeg (Wolny et al., 2020) and Stardist (Weigert et al., 2020), may help increase segmentation accuracy and overcome some of these bottlenecks. After segmentation steps, biosensor image analysis workflows also require additional bespoke steps, such as background subtraction and the calculation of emission ratios, to process the data, so it can be quantified meaningfully (Mahlandt & Goedhart, 2021; Rizza et al., 2017; Rowe et al., 2021; Waadt et al., 2020).

Without automation, the image analysis process can become laborious and one of the main bottlenecks in biosensor labs, taking much longer than the imaging itself. This bottleneck can be overcome with flexible software environments, such as Fiji or Icy, where custom workflows can be made, tested and applied quickly in batch (De Chaumont et al., 2012; Schindelin et al., 2012).

Combining high-resolution biosensor image sets with fully labelled and segmented image analysis tools may soon be widely achievable with use of deep learning tools. This would offer many exciting possibilities for future work, for example, correlating the size, position and growth rates of all cells within an organ with hormone concentrations, or examining how cellular organisation and connections affect hormone accumulation sites. These sorts of approaches have already yielded impressive results with the transcriptional and degron reporters. Models of root auxin flux based on cell geometries and transporter levels extracted from images have demonstrated the importance of plasmodesmatal connections to reproduce measured DII-VENUS patterns (Mellor et al., 2020). As the integration of hormone responses with precise spatial information has allowed transport models to recapitulate auxin response patterns, direct GEFBs will spur integrated models of *in planta* hormone biochemical pathways to dissect complex phenomena such as hormone distributions.

## 5. Breaking down biochemistry… and building it up

Biochemical pathways can be examined in models by representing component reactions numerically. A challenge is that parameters determining reaction rates are often difficult to directly measure. Statistical approaches are usually employed to explore the parameter space and identify which parameters have the greatest influence on model behaviour (Aldridge et al., 2006; Vernon et al.,

2018). Parameterisation is hampered by the difficulty of measuring the concentrations of individual chemical species, leaving many parameters unconstrained.

Biosensors allow the real-time quantification of metabolites, *in planta,* which greatly improves the parametrisation process. By using knockouts, applying reaction precursors and using inducible genetics, the relative contributions of various steps in a pathway can now be quantified. This approach was recently taken by Rizza et al. (2021), to characterise how a gibberellin gradient is established in the *Arabidopsis* root.

Previous multiscale modelling based on the expression of a Gibberellin 20-oxidase (GA20ox) biosynthesis gene had predicted that gibberellin levels would reach a maximum at the start of the root elongation zone and would diminish via dilution as cells elongated and progressed to the differentiation zone (Band et al., 2012). This contrasted with the gibberellin distribution observed with the gibberellin sensitive nlsGPS1 FRET biosensor (Rizza et al., 2017), which showed low gibberellin levels in the meristem that increased as the cells elongated. Therefore, Rizza et al. (2021) performed iterative gibberellin measurements, perturbations and mathematical modelling to determine that a gradual increase in gibberellin biosynthesis across the elongation zone best explains how the gibberellin distribution is generated.

Rather than the presumed rate limiting GA20ox enzyme step, both early biosynthetic steps and the final Gibberellin 3-oxidase (GA3ox) catalysed step were locally rate limiting in the elongation zone, the site of high gibberellin accumulation in *Arabidopsis* roots (Rizza et al., 2021). In contrast, GA20ox along with GA3ox biosynthetic steps were together rate limiting for gibberellin accumulation in the most apical meristematic region. Exogenously applied gibberellin reinforced the endogenous gradient, rather than eliminating it, a behaviour that was best reproduced in silico by assuming differential plasma membrane permeability to gibberellin (Rizza et al., 2021). The model predicted that artificially increasing permeability along the root would eliminate the gibberellin gradient when exogenous hormone is applied, which was tested in vivo by briefly applying gibberellin at low exogenous pH where indeed the gradient was greatly reduced.

This level of resolution when dissecting the functionally relevant steps of a biochemical pathway across an organ was made more accessible through the use of direct GEFBs and inspires further questions. Although the nlsGPS1 biosensor illustrates a correlation between gibberellin and cellular growth (Rizza et al., 2017), it is now clear from gibberellin perturbation experiments measuring gibberellin changes alongside growth that a simple dose-response model is insufficient to explain the relationship between the hormone and root cell behaviour (Rizza et al., 2021). This work demonstrates the power of the fast spatiotemporal responses of biosensors, in reliably quantifying metabolites and accurately parametrising biological models to move research forward quantitatively.

## 6. Focus on the future

Plant biosensors are already starting to move plant biology in a more quantitative direction. Complex metabolic processes are now being dissected at the level of spatially separate yet interconnected steps in vivo. A striking example is the recent characterisation of the photosynthesis dependent changes in NADP(H) and NAD(H) in chloroplasts, mitochondria, cytosol and peroxisomes using the iNAP and SoNar sensors (Lim et al., 2020). Using elegant combinations of biosensors, inhibitors and illumination treatments, this work demonstrates the flow of reducing equivalents between chloroplasts, peroxisomes and mitochondria through the cytoplasm to support photorespiration. The compartmentalisation of different biochemical processes into cells and organelles is an essential part of eukaryotic life, but it has historically proved challenging to study the interactions of these primary metabolic pathways *in situ*.

Targeting biosensors to different tissues and compartments like this may sometimes require some sensor re-engineering, but it is an approach that is already yielding great new insights into signalling, revealing the spatial specificity of responses. The targeting of R-GECO1.2 and G-GECO1.2 to the cytoplasm and nucleus demonstrated nuclear calcium spikes in *Arabidopsis* (Kelner et al., 2018; Leitão et al., 2019), and genetically disrupting these spikes with mutations in a nuclear cation transporter causes altered auxin homeostasis and root growth (Leitão et al., 2019).

Continued biosensor development may offer the tools to answer many of the big questions within plant biology, as new biosensors for difficult to measure metabolites, hormones, molecular events and even biophysical conditions are developed. Innovative new sensors, such as a recently developed crowding sensor for osmolarity (Cuevas-Velazquez et al., 2021), will allow the experimental integration of stress responses, biomechanics and gene expression.

The ongoing revolution in image analysis offers many exciting possibilities for future work with biosensors, through the correlation of spatial and tissue organisational data with biosensor outputs. High spatiotemporal resolution, nondestructive imaging in conjunction with comprehensive image analysis allows access to information that was unimaginable a few years ago. This approach was recently used to demonstrate a temporal centrifugal wave of auxin dynamics that precedes growth in the shoot apical meristem, by using degron-based auxin reporters (Galvan-Ampudia et al., 2020). Measuring hormone concentrations *in situ* has also improved models demonstrating which biochemical steps are essential to create and maintain hormone gradients (Rizza et al., 2021). These methods are broadly applicable, and the construction and parameterisation of a variety of biochemical and morphodynamic models will be improved with better data. This is perhaps the most exciting use of biosensor data—to close the loop between experimentation and mathematical modelling. The quantitative nature of biosensors offers many exciting future possibilities in plant research, and, though not yet measurable, the future seems bright.

## Acknowledgements

The authors would like to thank Martin Balcerowicz and Marino Exposito-Rodriguez for their helpful comments on the manuscript.

**Financial support.** The authors would like to thank their funders: the Gatsby Charitable Foundation and BBSRC (BB/P018572/1).

**Conflicts of interest.** The authors would like to declare they have no conflicts of interest.

**Authorship contributions.** JHR and AMJ wrote the manuscript. JHR made the figures.

**Data availability statement.** Code for the analysis presented in Figure 2, as well as test dataset, is available at https://github.com/JimageJ/ImageJ-Tools.

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
