## [Reviewer Report]

*Comments to Author*: The review by Rowe and Jones provides an overview of genetically encoded fluorescent biosensors (GEFBs) and recent advances in the field. The authors also provide examples of quantitative analysis using GEFBs and discuss well the problems of quantitative research. However, the reviewer unfortunately could not figure out from the text what the “two of the main bottlenecks” the authors mentioned in the abstract were. Also, the word “this” is often used throughout the text, but it is often unclear what it is referring to. Please revisit it. The figure is too small to see the text, and it is hard to understand.

There are also several minor points and editorial comments that may help to improve the clarity of this review further.

Page 2.

*“DII-Venus ubiquitination causes rapid degradation, reducing the time delay between hormone accumulation and fluorescence response.”*

The author later stated that these sensors cannot be used to image fast hormone depletion or oscillations, so it would be helpful if the author could provide the time required for these reactions.

*“A powerful tool to overcome this problem is the expression of a second fluorescent protein, as in the R2D2 or qDII reporters (Figure 1B).”*

These reporters are not described in Figure 1B.

*“Many also have a ratiometric output to control for optical artefacts during imaging and differences in expression.”*

I did not follow what optical artefacts mean. It would be helpful to the reader if the authors provide examples with appropriate papers.

Page 2. *“Herud-Sikimic et al., 2020”* bioRxiv paper was published in Nature 2021. The reference should be changed.

Page 3. *“Band, Úbeda-Tomás et al., 2012”* does not show the multiscale modeling based on the expression of the GA20ox biosynthesis genes. Please refer to the appropriate paper or rephrase the sentence.

*“Because this contrasted with the pattern observed with the nlsGPS1 FRET biosensor,”*

It is unclear what “the pattern” means. It would be clearer what the pattern is if the author explains what the nlsGPS1 FRET biosensor is. It would also make the text easier to understand if the author could be more specific about what “this” refers to.

*“The model predicted that increasing permeability along the root would eliminate the gradient when exogenous hormone is applied, which could be tested in vivo by applying gibberellin at low exogenous pH and indeed resulted in a loss of the gradient.”*

It would be helpful to the reader if the author could describe this in more detail. Why does the low exogenous pH increase the permeability of gibberellins along the root? Doesn’t the low pH affect the sensor itself?

Page 4.

*“Recent pioneering co-expression of pair of direct biosensors has allowed the temporal dynamics of many second messenger or hormones to be characterized simultaneously (Waadt et al., 2017, 2020). This has confirmed many predictions and offered new insight.”*

It would be better to write specifically about which second messenger and hormone dynamics are actually observable in the paper instead of using “many”, which predictions have been clarified by these technologies, and what new insights it has brought.

Minor edits:

Page 2.

mean → means

hormone depletions → hormone depletion

Page 3.

difficulty measuring → difficulty of measuring

real time → real-time

modelling determine → modelling to determine

---

## [Reviewer Report]

*Comments to Author*: Dear Authors

Thank you for submitting to QPB. We received the reviewers from two external reviewers, of whom one reviewer prefer not to reveal his/her identity. Both reviewers gave valuable and constructive advice on how to improve the quality of the manuscript further. 

Below are the comments of the anonymous reviewer:

The manuscript entitled „Focus on biosensors: looking through the lens of quantitative biology” by Rowe and Jones reviews, summarizes, highlights, some characteristics, applications, new aspects and future directions of genetically encoded biosensor research in plants. The manuscript is well written and apart from some small inaccuracies (mentioned below), I enjoyed reading it. I am missing a bit the concept of this article, because sometimes I had the feeling that the authors just throwed in some bullet points. The context and connections of the different sections was not clearly visible. Also, the discrimination between indirect transcriptional reporters and direct sensors was not precise. The terms extrinsic and intrinsic sensors and the explanation of this came too late. Maybe explain first the difference between indirect and direct and intrinsic and extrinsic biosensors, before discussing them in further detail.

Minor comments:

1) For helping the reviewer to properly comment on the article and the author to find the respective text, it is strongly recommended to also include line numbers to the manuscript.

2) “In recent years, plant biologists have started to transition from transcriptional reporter systems to genetically encoded fluorescent biosensors (GEFBs).” Transcriptional reporter systems, promoter:GFP reporters, also fall into the category of GEFBs. They are also genetically encoded. Please specify better or remove “transition from”.

3) “Biosensors allow the in vivo quantification of biochemistry…” Not all biosensors allow this. Please be more precise. Maybe use GEFBs here.

4) Although the abstract is clearly written, the text just provides some bullet points about Biosensors and GEFBs. The scope of this article is however not clear.

5) “These are proteins that contain one or more FPs, …”. The definition of GEFBs is a bit vague. Please be more precise.

6) “… see our recent review…” Remove “our”, as the first author of this article is not a co-author of this citation. Better stay neutral with such statements.

7) “From transcriptional reporters to genetically encoded biosensors”. Transcriptional reporters also belong to genetically encoded fluorescent biosensors. Maybe discriminate by direct and in-direct biosensors. Transcriptional reporters are indirect, FRET sensors are direct etc…

8) “… but at best, transcriptional reporters such as DR5 indicate where hormones were.”. These reporters do not detect the hormone itself, but the hormone response, so they indicate where the hormone response is strongest compared to other tissues.

9) “… due to changes in expression and degradation.”: Of what?

10) “By comparing the DII-nVenus fluorescence”. Please specify what the “n” stands for. Do you mean “m” for monomeric?

11) “…proteasomal machinery elicit fluorescence…” elicited?

12) “However, DII:Venus, R2D2 and qDII…” This sentence appears incomplete.

13) “… these are the direct GEFBs”. Good, so also define the indirect GEFBs above.

14) “…the first plant hormone GEFB were…” GEFBs?

15) “Herus-Sikimic et al., 2020” This is a not yet peer-reviewed article. Also, by looking at the properties of this GEFBs and the content of the manuscript, in my opinion this article is not yet worth to be cited.

16) “… from the donor CFP…” Better use here “CFP variant”, as CFP is not used in ABACUS1-2µ.

17) “Hanson et al., 2004”. Maybe include also a reference for the first use of roGFP in plants.

18) “nlsGPS1 FRET biosensor” Good to also state here that this sensor is GA responsive/sensitive.

19) mathematical modeling “to” determine…

20) “Rather than the presumes…” Make clear that you are speaking in this paragraph about GA in Arabidopsis roots.

21) “a” behaviour that could only be…

22) “…increasing permeability…” of what?

23) “… the gradient…” of what?

24) “…many biological processes involved…” “involve”?

25) “Huang et al., 2019” is relevant for guard cells, but used the biosensor developed by Waadt et al., 2017. In this context maybe worth to cite also Keinath et al., 2015.

26) “Leitao et al., 2019”. In this context maybe cite also Kelner er at., 2018. Leitao used the biosensors that were developed by Kelner.

27) “Exposito-Rodriguez et al., 2017”. Not sure about this reference. Experiments were done after transient expression in tobacco cells with sensors that not really work in transgentic Arabidopsis plants. For subcellular ROS/redox, better cite work from Markus Schwarzländer.

28) Regarding image analysis and data processing: Another bottleneck forgotten to be mentioned by the authors is focal drifts, or ROI drifts due to continuous plant growth during biosensor imaging. How to compensate for this?

29) At some point in the manuscript maybe include a small paragraph about light-sheet microscope-based biosensor imaging.

Figure 1) Add a title. If not cited in the main text, cite also the other biosensors mentioned in the figure. ABACUS topology is wrong (in c), see citation from original article Fig 3a:” The overall domain order of ABACUS1 is N-terminus–edCitrine–attB1–ABI1aid–L52–PYL1–attB2–edCerulean–C-terminus.”. If you don’t want to change the topology, you should indicate that the N-terminus is at the acceptor site and the C-terminus at the donor site.

Please revise your manuscript according to the advice of both reviewers. 

Yours sincerely 

Boon Leong LIM

---

## [Reviewer Report]

We are pleased to submit our revised manuscript for consideration after addressing the very helpful comments of the editor and reviewer.

---

## [Reviewer Report]

*Comments to Author*: I appreciate the responses and revised manuscript. The changes made by the author led to an improvement of this review.

Minor comments:

"Band, Úbeda-Tomás, et al., 2012" is not listed in the reference list.

Line 212: “difficulty measuring” difficulty of measuring?

Line 271: “recently was recently” Either recently should be removed.

Line 276: “This is perhaps” It may be unclear what This refers to.

---

## [Reviewer Report]

*Comments to Author*: Dear Authors 

Thank you for submitting to QPB. We received the reviewers from two external reviewers, of whom one reviewer prefer not to reveal his/her identity. Both reviewers suggested minor revision before acceptance.

Below are the comments of the anonymous reviewer:

The revised manuscript by Rowe and Jones entitled “Focus on biosensors: looking through the lens of quantitative biology”, has been significantly improved. The concept of analyzing GEFBs using state-of-the-art image analysis pipelines and feeding the data into computational models is recognizable and the manuscript is overall much better sorted.

There are a few minor things that came to my attention and that could be improved:

Line 48, 83, 123, 137 and several more) sort publications by publication date (oldest first) and alphabetically.

Line 51) This jump from “stress responses” to “FP maturation time” is very disturbing and somehow leads to a very abrupt topic change of this section. I recommend revising this section so that it has a clear focus. Biological findings made with transcriptional reporters, or low temporal resolution of transcriptional reporters due to slow maturation times and long half-lives of FPs.

Line 204) such as?

Line 271) “recently” doubled.

Line 302) where a Venus FP?

Line 320) threshold the filtered image

Reviewer 1

I appreciate the responses and revised manuscript. The changes made by the author led to an improvement of this review.

Minor comments:

"Band, Úbeda-Tomás, et al., 2012" is not listed in the reference list.

Line 212: “difficulty measuring” difficulty of measuring?

Line 271: “recently was recently” Either recently should be removed.

Line 276: “This is perhaps” It may be unclear what This refers to.

---

## [Reviewer Report]

We’re pleased to submit a final revision following acceptance for this invited commentary and thank the reviewers and editors for their helpful comments.

---

## [Reviewer Report]

*Comments to Author*: Dear Dr. Jones

We are pleased to inform you that your article "Focus on biosensors: looking through the lens of quantitative biology" has been accepted for publication in Quantitative Plant Biology. 

Thank you for choosing QPB to publish your work, we look forward to receiving further contributions from your research group in the future.

Yours sincerely

Boon Leong LIM

Associate Editor